# Microfluidic Synthesis and Analysis of Bioinspired Structures Based on CaCO_3_ for Potential Applications as Drug Delivery Carriers

**DOI:** 10.3390/pharmaceutics14010139

**Published:** 2022-01-07

**Authors:** Ekaterina V. Lengert, Daria B. Trushina, Mikhail Soldatov, Alexey V. Ermakov

**Affiliations:** 1Central Research Laboratory, Saratov State Medical University of V.I. Razumovsky, Ministry of Health of the Russian Federation, 410012 Saratov, Russia; Lengertkatrin@mail.ru; 2Biomedical Engineering Department, First Moscow State Medical University (Sechenov University), 119991 Moscow, Russia; trushina.d@mail.ru; 3Department of X-ray and Synchrotron Research, Federal Scientific Research Centre “Crystallography and Photonics” of Russian Academy of Sciences, 119333 Moscow, Russia; 4The Smart Materials Research Institute, Southern Federal University, Sladkova 178/24, 344090 Rostov-on-Don, Russia; mikhail.soldatov@gmail.com

**Keywords:** microfluidic chip, calcium carbonate, vaterite, core-shell structures

## Abstract

Naturally inspired biomaterials such as calcium carbonate, produced in biological systems under specific conditions, exhibit superior properties that are difficult to reproduce in a laboratory. The emergence of microfluidic technologies provides an effective approach for the synthesis of such materials, which increases the interest of researchers in the creation and investigation of crystallization processes. Besides accurate tuning of the synthesis parameters, microfluidic technologies also enable an analysis of the process in situ with a range of methods. Understanding the mechanisms behind the microfluidic biomineralization processes could open a venue for new strategies in the development of advanced materials. In this review, we summarize recent advances in microfluidic synthesis and analysis of CaCO_3_-based bioinspired nano- and microparticles as well as core-shell structures on its basis. Particular attention is given to the application of calcium carbonate particles for drug delivery.

## 1. Introduction

Recent years have shown considerable progress in the development of approaches for the synthesis of materials for drug delivery systems. Moreover, the demands of biomedicine have led to a current tendency towards the fabrication of drug delivery carriers with a complex structure. Much development has been driven by a need to perform synthesis of highly uniform and homogeneous nano- and microcarriers. Although notable progress has been made on the synthesis of such materials, the real biomedical challenges require a further breakthrough in the synthesis of complex and multicomponent carriers with high reproducibility and homogeneity to perform efficient therapy even on a single-cell level. One of the promising ways of overcoming these limitations is the application of microfluidic technology, which opens new routes for the synthesis of advanced nanostructured materials. The emerging interdisciplinary technologies based on microfluidics are at the forefront of the lab-on-chip approach as a unique way to produce nano- and microparticles. The microfluidic technique offers a variety of advanced features, such as high reproducibility, low batch-to-batch variation, fine control over particle characteristics, and an easy way to scale-up. Considerable progress in the synthesis of such systems has been achieved at the intersection of modern technologies and biomimetic approaches. The microfluidic approach has contributed greatly to the synthesis of a range of drug carriers, from natural macromolecules, synthetic polymers, inorganic, and hybrid nano- and microparticles [1,2,3,4]. Calcium carbonate (CaCO_3_) is among the potential materials for biomedicine that have attracted considerable attention in the context of microfluidics synthesis and study. It is one of the most widespread minerals that exhibit a range of highly important properties in vast fields, from bio-geochemical technologies to nanomedicine. Calcium carbonate is applied in many areas, including global CO_2_ exchange and industrial water treatment energy storage, and is employed as a core molecule in the formation of the shells and skeletons of organisms. Calcium carbonate has been employed in the production of cement, glasses, paints, plastics, rubbers, ceramics, and steel. It is also a key material in oil refining and iron ore purification. Although CaCO_3_ has been known for hundreds of years, one remarkable application was found recently and has been extensively developed over the past few decades. Biomedicine has employed calcium carbonate for a different purpose due to its unique mesoporous structure and stiffness accompanied by good biocompatibility [5,6].

Calcium carbonate exhibits complex polymorphic behavior, and each one of its polymorphic modifications has a variety of features that govern the application field. The typical crystallization pathway of calcium carbonate in the biomineralization procedure begins with the infiltration of ions from the surrounding medium by a matrix, followed by the formation of solid precursors composed of amorphous calcium carbonate. In the next step, precursor particles undergo a crystallization process resulting in polymorphic modifications in dependence on the external factors [7,8,9,10,11,12]. Calcium carbonate represents a promising drug delivery carrier due to its high porosity, loading capability, and ability to template formation of the polymeric shells. One of the most promising polymorph modifications of calcium carbonate is polycrystalline vaterite in the form of spherical mesoporous particles [13]. Vaterite particles of micron and submicron scale range have demonstrated a high drug loading capacity, biocompatibility, and long-term and safe storage of the loaded drugs [14,15]. In general, the synthesis of CaCO_3_ is a simple procedure of mixing salt solutions under agitation followed by aging, and it results in particles of 0.3 to 10 μm in diameter with a porosity of about 40% [16]. However, the reproducibility of the crystallization and homogeneity of the particles remain challenging, as their formation depends on many factors which can hardly be controlled [17]. At the same time, biomedical applications generally require particles of sub-micrometer size, for example, for active coating [18] or drug delivery [19], since this size ensures efficient penetration of the particles into tissues and internalization of the particles by cells. The loading of calcium carbonate particles with bioactive substances is mostly based on the co-precipitation procedure, which enables loading of low-molecular-weight compounds, polymers, and even nanoparticles with an average efficiency of about 2–4 wt.%. In general, loading efficacy can be tuned by controlling CaCO_3_ porosity, which has been shown to depend on the temperature of crystallization under supersaturated conditions, since the spherulitic crystal growth is accompanied by Ostwald ripening of the nanocrystallites [20]. Using this method, calcium carbonate has been loaded with a variety of functional species, including magnetite and silver nanoparticles, for implementing functions of targeting and probing by surface-enhance Raman signals [21]. Besides, recent progress in the development of CaCO_3_-based microcarriers has allowed for the design of systems for transdermal delivery of antimycotics [22], delivery of antimicrobials [23], multifunctional proteins [24], and enzymes [25]. Along with the strategies for loading the vaterite particles with a functional cargo, a trend in the synthesis of the core-shell structures is of great interest. A variety of CaCO_3_-templated microcarriers have been developed, including those using calcium carbonate as a functional core [26] or hollow capsules with the core to be removed [27,28,29]. However, further prospects for the application of calcium carbonate in the biomedical field are limited by technological factors: first of all, by the low reproducibility of the crystallization (size, shape, dispersity, polymorph composition, and porosity of the product). Additionally, the synthesis of nanosized carriers, which are highly demanded in biomedical applications, has turned out to be a significant challenge. Understanding the mechanisms that drive the formation of CaCO_3_ promises to result in the ability to produce tailor-made particles suitable for various biomedical applications. From our point of view, the latest progress in microfluidic devices opens up the possibility not only of analysis of calcium carbonate formation in situ, but also of a customized synthesis in a highly controlled manner from the beginning to the drug-loaded carrier with appropriate surface modification. Here, we review the results on microfluidic synthesis and analysis of the calcium carbonate particles themselves and core-shell structures on their base.

## 2. Microfluidic CaCO_3_ Synthesis

The first simplest microfluidic techniques for the synthesis of calcium carbonate particles employed continuous-flow mixing of reagents in T-shaped channels (Figure 1) [30,31]. Although polymorphic modifications of the resulting particles were limited by calcite with poor homogeneity, these studies demonstrated the possibility in principle for easy crystal formation within the channel. The main problem with this approach was the uncontrollable time of the reaction associated with continuous-flow reactors, which led to broad size distributions of the resulting crystals and calcite polymorph modification.

More advanced microfluidic techniques for the synthesis of CaCO_3_ are based on laminar Y-shaped channels, which provide a mixing of the salts followed by aging, with ions movement governed only by diffusion (Figure 2) [32]. This effect is referred mainly to the laminar flow and low Reynolds number, which are the main parameters of microfluidic devices that determine chemical processes in the channel.

Simulation of the fluid dynamics revealed the supersaturation ratio in the CO_3_^2−^ side of the channel to be larger in comparison with Ca^2+^ ions due to deference in diffusion coefficients. This results in the tendency of CaCO_3_ formation at the CO_3_^2−^ side [32]. Evidently, this effect prevails at low rates of flow, while at high rates the crystal formation mainly occurs on the contact interface.

The crystalline phase, in the form of either calcite or aragonite, forms as a result of dissolution–recrystallization of the amorphous calcium carbonate that acts as a transient precursor [33,34]. Investigation of the process in the microfluidic channel revealed an important dependence of polymorph CaCO_3_ modifications on the concentration of the salts. In particular, the concentrations of reactants above 6 mM corresponded to the critical supersaturation and resulted in the rapid crystallization of calcite, while concentrations below 6 mM led to the formation of metastable amorphous calcium carbonate followed by the formation of crystalline vaterite-CaCO_3_ after a period of crystallization induction time [32]. Synthesis of aragonite modification was achieved by the application of a continuous microfluidic system with a complex structure of a “Christmas tree”-like configuration, which provides a gradient of reagent concentrations [35]. Laminar streams of solutions containing calcium and magnesium ions flowed in the channel with different concentration gradients, and it was found that the initial crystallization time increased exponentially and that the crystal density decreased with increasing concentration of Mg^2+^ ions. This result indicates the importance of additives in controlling the CaCO_3_ formation process. All precipitated particles were of a snowman-like or spherical shape; they were later transformed into a spinose sphere-shaped crystal, which was their final shape in this study.

Detailed investigation of the amorphous phase formation revealed additional metastable amorphous modifications. After mixing the reactant solutions, the precipitation of the initial branched aggregate of amorphous calcium carbonate (ACC) (I) was first observed (Figure 3). This was followed by the formation of a more ordered whisker-like ACC (II) phase based on the originally branched aggregates as a result of their dehydration and aggregation [36]. At the next stage, the complex behavior of ACC (II) occured, leading either to its gradual transformation into spherical crystallites and the subsequent vaterite formation through spherulitic growth, or to rhombohedral calcite as a result of the well-known dissolution–recrystallization process. In addition to the obvious dependence of these processes on the concentration of the reagents, the recrystallization path was also found to be pH-dependent [37] and size-dependent [38,39].

The amount of water contained in the particles is another important characteristic of amorphous calcium carbonate, which determines the evolution of its structure and composition in the early stages of crystallization in microfluidic devices [33,40]. Huachuan Du et al. employed a microfluidic spray-dryer device to reveal a strong correlation between the amount of mobile water contained in the ACC particles and the formation time, which means an increase in the amount of mobile water with increasing particle size. This effect has been demonstrated using formation times from 100 ms to 10 s with corresponding particles sizes of 30 nm to 130 nm [41]. In particular, an increase in the content of mobile water in calcium carbonate particles with a diameter of 130 nm by 30%, in comparison with particles with a diameter of 30 nm, leads to a significantly reduced thermal stability with respect to solid crystallization.

Biomineralization has potential for various engineering applications, e.g., improving soil strength, reducing the hydraulic conductivity of soils, immobilizing groundwater contaminants, enhancing oil recovery, increasing storage security of CO_2_, and fugitive dust control [42]. Biomineralization can be stimulated through microbially or enzymatically induced carbonate precipitation by stimulating urea hydrolysis in presence of dissolved calcium ions [42,43,44]. For this study, a special microfluidic system has been developed. Microfluidic CaCO_3_ precipitation was studied in the porous medium (soil particles) containing bacterial cells and a cementation solution (urea and calcium chloride) [43,44]. The overall microbial-induced CaCO_3_ precipitation processes from the beginning up to 12 h after the mixing of the bacterial suspension and the cementation solution can be divided into the following three main stages: (1) bacterial aggregation, which occured immediately after the mixing of the bacterial suspension with the cementation solution; (2) growth of irregularly shaped CaCO_3_ precipitates (0–1 h); and (3) dissolution of irregularly shaped CaCO_3_ precipitates (1–2 h) at the expense of the growth and formation of regularly shaped CaCO_3_ crystals (1–12 h). In general, the observable process is consistent with classical nucleation and growth theory and with Ostwald’s step law. There is an assumption that bacterial cell walls are negatively charged and can adsorb calcium ions from the environment, and that, therefore, once the bacterial cells hydrolyze urea and produce carbonate ions, they precipitate with the calcium cations attached to the bacterial cell walls. In this way, bacterial cells have been shown to serve as CaCO_3_ nucleation centers, so that CaCO_3_ crystals precipitate and continue growing around bacterial aggregates [44]. The irregularly shaped CaCO_3_ precipitates continued growing for approximately 1 h, and after that, the irregularly shaped CaCO_3_ that was smaller than a certain critical size started dissolving and the bacterial cells became free to move again. Between 2 and 12 h, all of the existing CaCO_3_ precipitates were formed as regularly shaped crystals. It was established that an injection time interval of 3–5 h is beneficial for the production of 5–10 μm particles, while at a longer injection interval (23–25 h) the crystals were larger—10–80 μm. Unfortunately, there is no information about the polymorphic composition CaCO_3_ and recrystallization during the dissolution–precipitation process, but the shapes of the crystals were found to be either spherical or prismatic, which might represent vaterite or calcite, respectively [43]. Importantly, it is not only the polymorphic modification that affects the loading efficiency of CaCO_3_; the shape and morphology of amorphous calcium carbonate also affect both the loading capacity and the activity of the entrapped therapeutic [45]. In particular, elliptical particles exhibit the highest loading efficiency in comparison with the spherical carriers. The size of the particles was also found to affect the encapsulation capability: smaller particles exhibited higher loading efficiency in comparison with larger ones. Star-like particles were found to be advantageous due to the large surface area, which can be employed for adsorbing large molecules onto the surface [46]. In this way, star-like particles had only a slightly higher loading capacity in comparison to spherical ones but exhibited a significantly higher activity of the encapsulated therapeutics [45]. Furthermore, the therapeutic activity increased with an increase in the loading capacity of the CaCO_3_ particles.

It is known that a small number of additives can significantly inhibit both the nucleation and the growth of calcium carbonate crystals; however, the presence of organic compounds in the media can inhibit growth, while favoring nucleation [42]. Proteins (enzymes) may also act as chelating compounds, lowering the concentration of free calcium ions available for precipitation. The crystal size typically demonstrates an exponential distribution, which gradually changes with multiple cycles of treatment (10 times of infiltration with a reactive solution) in a patterned microfluidic device. At the first cycle of treatment, the range of crystal size was relatively narrow (15–105 μm). Upon subsequent flushes of the reactant solutions to the microfluidic system, variation in crystal size became wider, reaching 20–455 μm at the ninth cycle of treatment. Besides an increase in the size, the number of crystals also increased with multiple cycles of treatment, which implies that nucleation of new crystals and growth of existing minerals took place simultaneously throughout each cycle. The number of crystals within the range of 20–60 μm increased most significantly from the first cycle to the fifth cycle, which confirms that the nucleation of new crystals is most dominant within the first five cycles of treatment. The volume of crystals gradually increased until about 14 h, after which time no further significant change in crystal volume was observed [42]. Although the naturally driven system excites interest, the results presented are poor in terms of homogeneity and reproducibility.

These results mean that calcium carbonate exhibits complex behavior from the nucleation to the crystalline phase and depends on many factors, some of which are interdependent. However, naturally derived CaCO_3_ exhibits superior properties desired in artificial synthesis, in this regard mimicking natural presses, and is of high interest in the synthesis of calcium carbonate particles. One of the natural conditions is the formation of the particles in confinement, represented either by a confined volume in a polymer matrix or by lipid bilayer vesicles. In biomineralization processes, amorphous particles act as precursor particles that are encapsulated within the vesicles, stabilizing and transporting them to the site of crystallization [7,9,47]. Confined conditions also ensure the accumulation of high concentrations of calcium ions without cytotoxic effects. In this way, particles of high reproducibility and homogeneity can be formed in such a way as to have properties unobtainable in other ways, such as curved surfaces [7]. A range of works have demonstrated efficiency of methods based on application of picoliter droplets [48], micropores [49], and polymersomes in the synthesis [50]. These studies show artificial confinements to efficiently guide and control the calcium carbonate crystallization process and polymorphism.

In this regard, biomineralization-mimicking procedures using droplet-based microfluidics were developed to produce amorphous calcium carbonate particles within fluid-phase unilamellar vesicles. In this case, the vesicles frame a confined volume for amorphous particle growth limited by the size of the vesicle, which can be varied from tens and hundreds of nanometers to tens of micrometers. To address these biomimetic ideas, Hannes Witt et al. developed a microfluidic system for the synthesis and analysis of calcium carbonate inside giant vesicles (Figure 4). This technique allowed reaching high concentrations of CaCl_2_ (100 mM), which cannot be easily achieved, and demonstrated calcium carbonate particles formed in the confinement to be in conformal contact with the lipid bilayer. Thus, strong interaction and colocalization between the particles and the membrane were suggested [51]. Detailed analysis of the CaCO_3_ crystals’ nucleation and growth in the confinement was performed by Jack Cavanaugh et al. using microfluidic polydimethylsiloxane (PDMS) chambers and liposomes, and revealed a steady-state nucleation rate of 1.2 cm^−3^·s for the crystal nucleation rate [52].

Another possibility for avoiding the limitations in the artificial production of CaCO_3_ is the adoption of a segmented flow regime employing reagents encapsulated within a large number of small identical picoliter droplets that move at a constant linear rate. These droplets represent artificial confinements that can be obtained using microfluidic water-in-oil emulsions, which provide nucleation and growth of CaCO_3_ within droplets (Figure 5) [53,54]. For example, CaCO_3_ nucleation kinetics was studied during 10,000 simultaneous reactions in identical droplets within a multilayer microfluidic chip [54]. This study revealed the nucleation to follow a double exponential function and to be 20 times in magnitude slower in droplets in comparison with bulk conditions. In contrast to droplet-based systems and vesicles, this approach does not require the application of surfactants, and allows the implementation of long-term processes such as slow nucleation.

Polymorph compositions of the resulting calcium carbonate were shown to be strongly dependent on the concentration of salts in the droplets: a concentration of 4 mM reagents in 250 pL droplets uniquely resulted in the formation of calcite, while 8 mM reagents enabled the formation of vaterite and the reaction in bulk conditions always resulted in the mixture of polymorphs in a different ratio [53].

The segmented-flow approach allowed the application of synchrotron X-ray techniques to study CaCO_3_ crystallization processes (Figure 6) [55]. M.A. Levenstein et al. described a versatile and re-usable microfluidic platform to study crystallization processes using synchrotron X-ray techniques, and their reported results demonstrated bioactive glass and NX illite to be effective nucleating agents for the synthesis of calcite, thereby offering a tool for control over the reaction within droplets in the channel. The droplet-based approach in segmented-flow conditions suggests minimization of the influence of impurities on the process of nucleation and crystallization, which prevail in a bulk solution. Droplet microfluidic-coupled X-ray diffraction enables the collection of time-resolved, serial diffraction patterns from a stream of flowing droplets containing growing crystals.

An interesting approach was reported by Seung Goo Lee et al., who developed a system with dynamically tunable geometry, porosity, and wettability. They employed a photolithographic technique accompanied by site-selective mineralization that allows mimicking of real carbonate reservoir properties throughout heterogeneous geometries [56]. Specifically, the authors controlled CaCO_3_ growth and dynamically adjusted the structure’s geometry by flowing Ca^2+^ and CO_3_^2−^ ions in rich/supersaturated solutions. Once the initial structure of CaCO_3_ particles is formed, the geochemistry of their surface can be altered by the flow of various fluids, including oil, water, CO_2_, and acids. Duy Le-Anh et al. showed an improved oil recovery from CaCO_3_ with the help of a newly developed microfluidic chip containing a 3D-packed bed of calcite particles [57]. They demonstrated no significant dependence of calcium carbonate loading with oil on the capillary number. Nevertheless, a chemical modification of the pore space via adsorption of water-extracted crude oil components led to significantly higher loading values, which indicates a good potential for using packed beds on a chip as an efficient screening tool for the optimization and development of different oil recovery methods. It should be noted that the porosity of calcium carbonate microparticles is an important parameter in biomedical applications, as it affects the encapsulation processes of biomolecules [58,59] and nanoparticles [60], and the porosity affects the processes of core-shell particle fabrication [61,62].

The small volume and the identity of the droplets allow for synthesis of the particles with a high reproducibility and homogeneity, as well as for carrying out the crystallization without convection and contamination, which are typical for the processes in bulk conditions. Combined with several analytical methods, the droplet-based approach has been successfully employed to synthesize high-quality CaCO_3_ particles with precise control over the nucleation kinetics, polymorphism, and particle properties.

## 3. CaCO_3_-Based Core-Shell Structures

Calcium carbonate has several properties that set it apart from many other minerals. Besides high biocompatibility, CaCO_3_ has a mesoporous structure capable of loading with different cargo. In addition to the surface modification of CaCO_3_ by different polymers, CaCO_3_ demonstrates unique results for biomedical applications, e.g., in bone tissue engineering. For this, P. Xu et al. synthesized CaCO_3_/gelatine methacrylate microspheres in a one-step in situ process using a non-planar flow-focusing microfluidic device, and demonstrated that human umbilical vein endothelial cells and immortalized mouse embryonic fibroblasts can easily attach and adhere to the surface of these microspheres and maintain high viability [63].

The in situ gelation process in combination with microfluidic technology offers the production of highly homogeneous microcarriers with precisely tunable size and structure, and allows for control over the crosslinking process. Usually, alginate gel can be cross-linked by two main approaches, the first of which suggests the application of divalent cations such as Ca^2+^, Sr^2+^, and Ba^2+^, or by covalent chemical reactions [64]. Besides, alginate is a thermally stable polymer that undergoes a rapid gelation process and is able to form hydrogels at room temperature [65]. However, traditional methods to produce alginate microstructures most commonly lead to a high level of polydispersity, which significantly limits potential applications in biomedicine. It should be noted that carriers of smaller size possess a larger surface-area-to-mass ratio and, as a result, exhibit higher encapsulation efficiency [66,67].

A promising application of CaCO_3_ was found in combination with sodium alginate, which undergoes gelation under the influence of Ca^2+^ ions. In this regard, a variety of configurations were investigated for employing calcium carbonate as both a functional core and a donor of Ca^2+^ ions to induce the gelation process. Monodispersed spherical calcium alginate microgels with uniform and controllable sizes from 40 to 700 μm were prepared using a microfluidics technique [68]. In comparison with conventional external crosslinking, this method can avoid the deformation of microgels. Alginate hydrogels are the promising material for the production of microcarriers for delivery systems, which enable encapsulation of both proteins and substances of low molecular weight, accompanied by the possibility of controlled release. Alginate microgel exhibits excellent biocompatibility, biodegradability, and mild gelation process. Furthermore, such spherical calcium alginate microgels can improve their movement and packing state in a microchannel, which is beneficial for their use as embolic materials for interventional therapy.

Application of CaCO_3_ as a Ca^2+^ depo was employed to produce spherical and bullet-like alginate microcapsules with a core-shell structure by the deformation of double-emulsion droplets [69]. In this regard, the water phase was filled with the mixture of alginate and calcium carbonate particles followed by treatment with acetic acid, which led to the burst release of Ca^2+^ ions and the gelation of alginate (Figure 7).

Spontaneous release from these spherical and bullet-like capsules was tested using the antioxidant food additive α-tocopherol. The results show that the bullet-like microcapsules with a single inner core exhibit a faster release rate than do spherical microcapsules in a phosphate buffer solution. Remarkably, the shape of the shell and the number of cores were shown to affect the kinetics of the release. A thinner local alginate shell and a high surface area are considered to be among the reasons for this phenomenon, based on the investigations on the release mechanism of alginate microcapsules [70]. These point to the release from alginate shell as being driven by three processes: burst release, swelling and erosion release, and diffusion release. Nevertheless, carrier erosion is considered the predominant mechanism of drug release from CaCO_3_-based alginate microcapsules [69].

The shape of the particles is an important parameter in drug delivery applications, as it was shown to affect intracellular uptake [71]. The application of anisotropic particles as a template in the synthesis of capsules enables the obtaining of capsules characterized by anisotropic properties [72], which is essential in a range of applications [73], such as oxygen delivery [74], shape-controlled cellular uptake [75,76], or flow dynamics [77].

The authors of another study improved the microfluidic approach to increase the barrier properties of the CaCO_3_-templated alginate shells by coating the alginate shells with a layer of oppositely charged polymer (polyethyleneimine and chitosan), which can act as a diffusion barrier [78]. This study showed an encapsulation efficiency of up to 90%, although the spherical shape of the alginate shells was achieved by finishing the formation of the shells in the bulk conditions. A recent study has shown the synthesis of the monodisperse alginate shells by templating the formation with CaCO_3_ particles in confined volume, following the Ca^2+^-assisted crosslinking procedure described above [79]. Low pressure (400 mbar) and correct surfactant concentration were shown to be crucial for the synthesis of monodisperse core-shell structures with a small diameter and high stability.

## 4. Outlook

A number of chemical approaches for CaCO_3_ particle synthesis and surface modification have been developed in recent decades. While it might be challenging to predict the future developments in CaCO_3_ nano- and microparticles synthesis, the general trend in state-of-the-art science and technologies is evident. Artificial intelligence (AI) approaches are the next round in developing the solution for practical problems in science and technologies, including the development of new materials. Such technologies go far beyond the borders of computer science and provide new insights to distant areas of expertise with chemical applications. One of the attractive applications is an automated material-synthesis laboratory driven by AI. The introduction of high-level laboratory automation and the AI approaches to microfluidics will determine the future of the synthesis of new materials and the optimization of known reactions [80]. We believe that the combination of in situ synchrotron techniques [55] with AI synthesis in microfluidic systems will determine the development of CaCO_3_ nano- and microparticles synthesis, surface modification, and even drug delivery applications.

## 5. Conclusions

In conclusion, microfluidics offers a wide variety of tools and new technological opportunities for implementing physical and chemical processes in a highly controlled manner accompanied by in situ analysis of the reactions. Several studies have shown the possibility in principle of the microfluidic synthesis of calcium carbonate particles to enable control over a range of properties such as size, shape, porosity, polymorphic modification, and others, which are crucial for biomedical applications. Great progress has been achieved by the application of biomimetic approaches in microfluidic devices, such as synthesis in confinement and, in some cases, synthesis mediated by an organic matrix or by modulation of the concentration of the reactants in time. Confinements represented by either vesicles or water-in-oil droplets suggest minimization of the influence of impurities on the nucleation and crystallization process that dominates in bulk conditions. The small volume and identity of the droplets allow for the synthesis of CaCO_3_ particles with high reproducibility and homogeneity, as well as for carrying out crystallization without any contamination. Microfluidic systems are also beginning to be explored as platforms for the development of more complex core-shell structures, where CaCO_3_ often serves as a core material. Indeed, microfluidic devices allow for the effective generation of alginate microparticles and microcapsules while controlling the morphology and chemical properties to improve the physical properties, solubility, and biocompatibility of alginate, accompanied by the ability to store and deliver drugs. However, progress in the synthesis of core-shell structures is still far from ideal. In general, the microfluidic approach enables precise control over the shape of CaCO_3_ particles and its morphology, which in turn affects the activity of encapsulated therapeutics.

Microfluidic synthesis of functional core-shell structures, including those based on CaCO_3_, is a poorly developed area that has prospects for further progress. The first works on the microfluidic synthesis of calcium carbonate were published more than 10 years ago; however, templating of functional core-shell structures has been shown only just recently. Most of the studies have been conducted from the fundamental point of view either of estimating the processes of CaCO_3_ formation or of showing the possibilities in principle of its templating. However, these studies reveal crucial parameters that offer new knowledge and possibilities for improving current techniques, and open up a venue for further progress in the development of drug delivery systems.

## Figures and Tables

**Figure 1 pharmaceutics-14-00139-f001:**
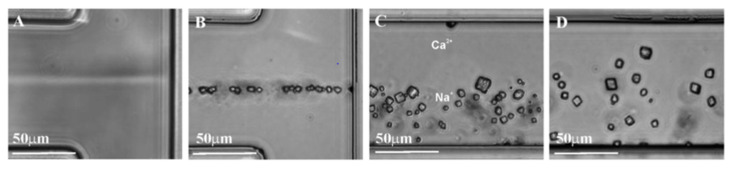
CaCO_3_ crystal formation in laminar flow after mixing Na_2_CO_3_ and CaCl_2_ solutions in deionized water: (**A**) initial flow cell; (**B**) formation of the crystals at the interface of the salts; (**C**) crystal formation on the CO_3_^2−^ side of the stream; and (**D**) uniform distribution of crystals over that laminar flow. Image reproduced from Huabing Yin et al., with permission from Anal. Chem. [30].

**Figure 2 pharmaceutics-14-00139-f002:**
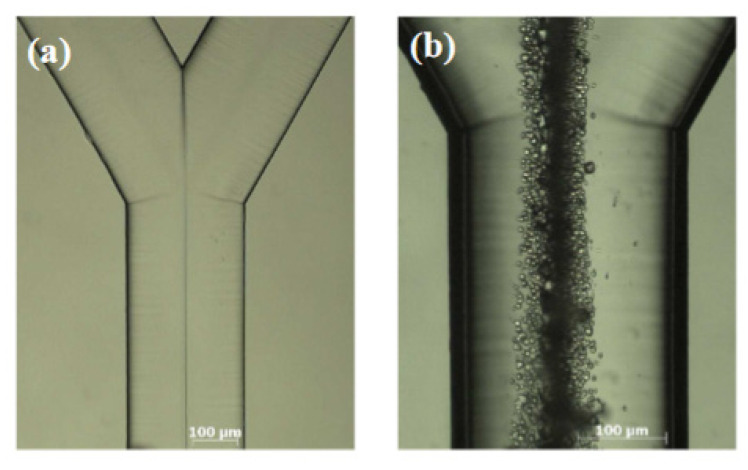
(**a**) Interfaces between the salt solutions in laminar flow and (**b**) CaCO_3_ formation at the interface of the salts. Image reproduced from Youpeng Zeng et al., with permission from Cryst. Growth Des. [32].

**Figure 3 pharmaceutics-14-00139-f003:**
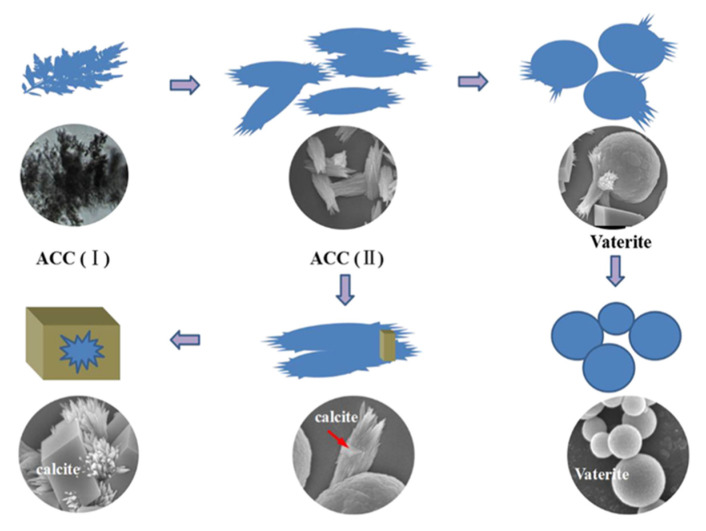
Schematic image of the crystallization pathway of amorphous and crystalline CaCO_3_ and corresponding SEM images. Image reproduced from Youpeng Zeng et al., with permission from Cryst. Growth Des. [36].

**Figure 4 pharmaceutics-14-00139-f004:**
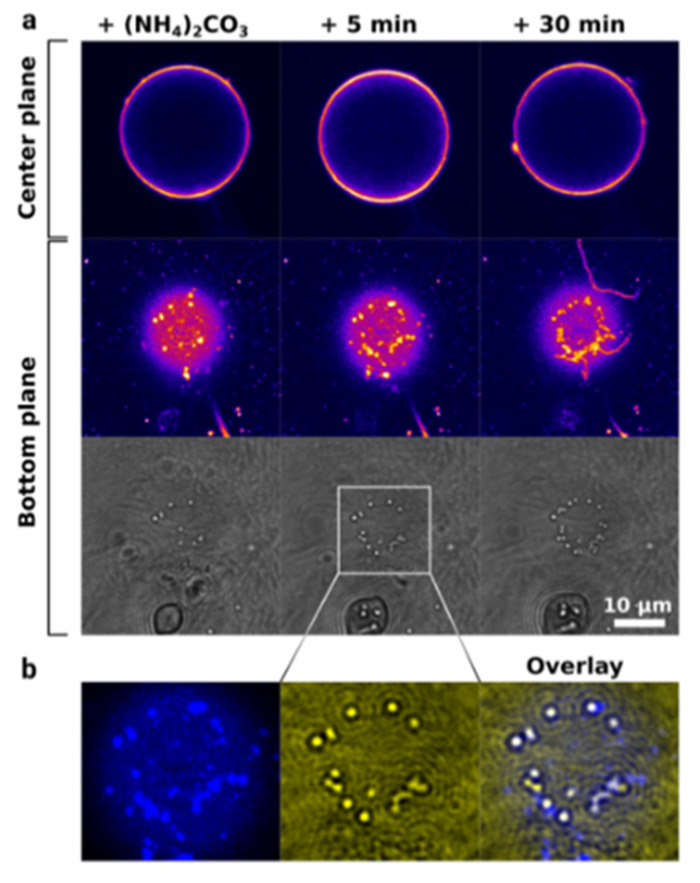
Crystallization of calcium carbonate within a vesicle: (**a**) CLSM images of a labeled vesicle at the central plane and bottom plane, and corresponding bright-field images at the bottom plane. (**b**) Magnified CLSM image of the labeled vesicle (left), bright-field images of CaCO_3_ particles at the central place, and an overlay of the two images. White spots in the overlay correspond to the colocalization of CaCO_3_ and the lipid membrane of the vesicle. Image reproduced from Hannes Witt et al., with permission from Langmuir [51].

**Figure 5 pharmaceutics-14-00139-f005:**
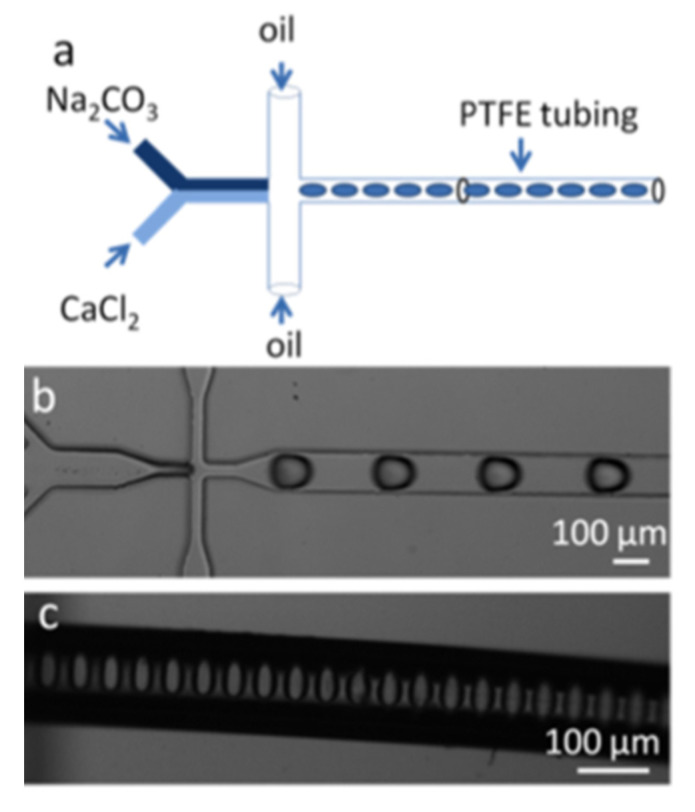
(**a**) Schematic of the Y-shaped microfluidic system used for the segmented flow synthesis of calcium carbonate. Droplets are generated by an FC-40 oil flow from the additional channels. (**b**) Bright-field Image of droplets. (**c**) Image of aqueous droplets of calcium carbonate. Image reproduced from Yashina, A. et al., with permission from Biomicrofluidics [53].

**Figure 6 pharmaceutics-14-00139-f006:**
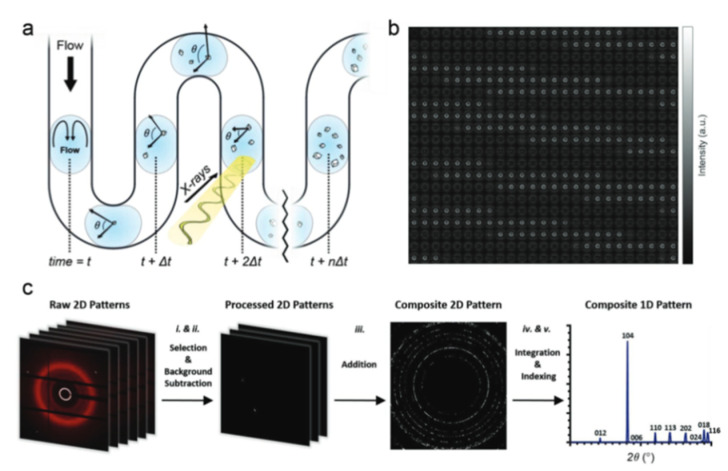
(**a**) Schematic image of the experimental set-up with an X-ray beam held at the fixed position. (**b**) Images of 500 consecutive diffraction patterns of the droplets during the calcite particles formation. (**c**) Diagram of the 2d patterns processing and analysis; (**i**) frames containing oil are discarded; (**ii**) background are subtracted from the selected frames; (**iii**) processed frames are combined to obtain a composite diffraction pattern; (**iv**,**v**) the composite pattern is integrated to obtain the diffraction profile. Image reproduced from Levenstein, M. A. et al., with permission from Adv. Funct. Mater. [55].

**Figure 7 pharmaceutics-14-00139-f007:**
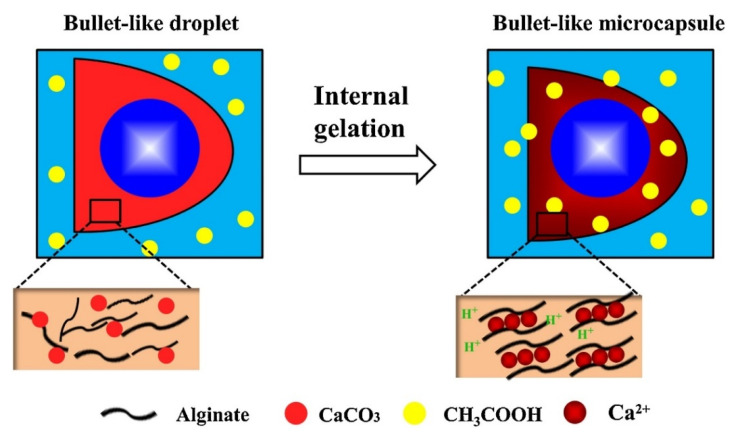
Schematic picture of the Ca-alginate core-shell microcapsule formation by means of internal gelation in monodisperse double emulsions. Oil/Water/Oil droplet templates were obtained via a dual-coaxial capillary microfluidic device. O/W/O droplets were produced at the second stage by the shearing force of the outer oil phase. Image reproduced from Huang, L. et al., with permission from Colloids Surf. A Physicochem. Eng. Asp. [69].

## Data Availability

Not applicable.

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
