# Peer review of "Microfluidic Synthesis and Analysis of Bioinspired Structures Based on CaCO3 for Potential Applications as Drug Delivery Carriers"

_pharmaceutics, 2022, doi:10.3390/pharmaceutics14010139_

Round 1
Reviewer 1 Report
Probably the picture of ACC (2) is not ACC. The particle size is not written, but I think it is rather large. The particle size of ACC should be finer.
Author Response
We thank reviewer for this comment, indeed, the ACC(2) particles seem to be rather large, however authors [Youpeng Zeng et al. Cryst. Growth Des. 2018, 18, 11, 6538–6546 10.1021/acs.cgd.8b00675] persuasively proves these particles to be amorphous phase rather than calcite as it could be concluded from the SEM images.
Reviewer 2 Report
The article presented by Lengert and coworkers describes some methods for the synthesis and analysis of CaCO3 and CaCO3- based particles. In general, the article is well written and presents interesting insights on microfluidic methods, as well as on the different approaches that can be applied to control both the polymorphism and the particles size. Interesting and recent references to the potential drug delivery applications of CaCO3 particles are reported in the introduction: in the following sections, few more examples of drug delivery and biomedical applications of micro/nanoparticles produced with the methods described should be added.
The experiment reported in lines 206-222 should be explained better, as the reactants, the microfluidic device geometry and the “multiple cycles” approach choice are not clearly described. Either the reference to the related work is missing, or, if it is an experiment performed by the authors, they should explain it better.
Figure 7 is not clearly explained: a few lines should be added to describe briefly the setup used to produce the droplets.
Once these few issues are addressed, the paper should be accepted.
Typos and other comments:
On line 103, begging – is it supposed to be beginning?
On line 327, a Figure 9 is cited, but this figure is not present in the manuscript.
Author Response
Reviewer: The article presented by Lengert and coworkers describes some methods for the synthesis and analysis of CaCO3 and CaCO3- based particles. In general, the article is well written and presents interesting insights on microfluidic methods, as well as on the different approaches that can be applied to control both the polymorphism and the particles size. Interesting and recent references to the potential drug delivery applications of CaCO3 particles are reported in the introduction: in the following sections, few more examples of drug delivery and biomedical applications of micro/nanoparticles produced with the methods described should be added.
Answer: We agree with this comment. A few more paragraphs are now added over the manuscript to show the importance of the results described in the review in terms of drug delivery demands. (page 6 lines 206-216, page 9 lines 328-331, page 11 lines 395-400)
Reviewer: The experiment reported in lines 206-222 should be explained better, as the reactants, the microfluidic device geometry and the “multiple cycles” approach choice are not clearly described. Either the reference to the related work is missing, or, if it is an experiment performed by the authors, they should explain it better.
Answer: We thank the reviewer for this comment. Indeed, the reference has been missing in this paragraph, we add the reference and some additional explanations to the manuscript. (page 6, lines 222-223)
Reviewer: Figure 7 is not clearly explained: a few lines should be added to describe briefly the setup used to produce the droplets.
Answer: We add few important details in the description of the Figure 7
Reviewer: Once these few issues are addressed, the paper should be accepted.
Reviewer: Typos and other comments:
Reviewer: On line 103, begging – is it supposed to be beginning?
Answer: Yes, sure it is typo which is corrected now, thank you for the comment.
Reviewer: On line 327, a Figure 9 is cited, but this figure is not present in the manuscript.
Answer: A couple of figures were removed from the final manuscript which led to this mistake, it is corrected now
Reviewer 3 Report
Thank you for providing such an informative review on recent advances in microfluidic synthesis and analysis of CaCO3-based nano-and microparticles as well as core-shells structures on its basis. In my opinion, it is a well-organized text and the distribution of references was good. however, some minor grammatical points need to be checked before online publishing.
Author Response
We thank the reviewer for the opinion and time. We made additional editing and corrections over the text to improve the manuscript.
Reviewer 4 Report
This review by Ermakov and collaborator is a nice account on the state of the art of the synthesis of nano and micro particles of calcium carbonate. In particular, the preparation of such materials using microfluidic techniques. However, the number of applications discussed in text of these materials in drug delivery is vey limited. In lines 57-58: “Bio-medicine has employed calcium carbonate for a different purpose due to its unique mesoporous structure and stiffness accompanied by good biocompatibility.” The authors add only two references (5,6). Hence, I feel that the topic of this work is not the applications but the technique of preparation of these materials. I would like to ask the author to provide additional references on the applications and discuss the resulting systems.
Author Response
We thank the reviewer for the important notice on the discussion of applications. We agree with this comment and we add additional information on the practical importance of the described progress over the manuscript. (page 6 lines 206-216, page 9 lines 328-331, page 11 lines 395-400)
Round 2
Reviewer 4 Report
The authors have introduced some new information about the applications of CaCO3 for drug delivery applications as was requested.